# Learning Robust Representations for Medical Images via Unifying (Self-)Supervisions

## Abstract

Pre-training medical image encoder to provide robust, task-agnostic representations is highly valuable, as it enhances the understanding of medical images and is important for performing many data-scarce analysis tasks. Current pre-training works are unable to integrate various types of supervisions, including self-supervision and external supervision such as segmentation annotations, while they are highly valuable for medical image understanding. Therefore, in this paper, we take the first step toward exploring unifying all common types of supervisions into a pre-training framework through a same scalable way. This require the pre-training framework being both *unified*, for accommodating diverse data and extensible, and *effective*, for making heterogeneous data synergistically assist unknown downstream tasks. To this end, we propose *UmiF*, whose principle is that once converted into token embeddings in a unified space, all diverse supervisions can be effectively utilized via contrastive learning and mask modeling with a same way. With *UmiF*, we pre-train on 1.66M samples from 14 public datasets, significantly surpassing previous efforts in terms of the dataset scale. We obtain and release [1] the *UmiF* model, which achieved state-of-the-art performance across various downstream tasks, including classification, segmentation, and detection, retrieval and VQA.

## 1 Introduction

As a practical application field, medical image analysis tasks are highly diverse, including diagnosis, prognosis and progression prediction for different diseases, as well as segmentation for organs or lesions. Despite recent advancements in deep learning (He et al., 2016; Dosovitskiy et al., 2020b), many critical practical problems lack sufficient data for training a deep model. For instance, pediatric interstitial lung disease (Guillerman, 2010), primarily affects children and is rare, resulting in insufficient high-quality CT data to train a robust deep model. One promising direction is *pre-training task-agnostic medical image representations* on large datasets with general learning objectives. Such representations provide basic understandings to medical images, and can achieve better performances on downstream tasks via further fine-tuning or even zero-shot adaptation (Qiu et al., 2023).

Many previous works explore pre-training techniques in medical images. In terms of the supervision type, they can be generally divided into two groups. One group of works mainly use language as supervisions to guide image representation learning (Zhao et al., 2023; Shrestha et al., 2023). These pre-trained models focus on image-level downstream tasks like classification while fine-grained patch-level information is not emphasized. The other group of works use supervisions in images themselves and employ self-supervised learning methods like DINO (Pérez-García et al., 2024) and MAE (Zhou et al., 2023b). However, some supervisions, such as paired texts, segmentation annotations and classification labels, are largely overlooked by them in the pre-training stage. These labels often come from doctors with rich domain knowledge, incurring extremely high costs and possessing significant value on medical image understanding and high-quality visual features. Some works are also attempting to combine different training signals, such as incorporating labels within the vision-language pre-training framework (Wu et al., 2023). However, these efforts are mostly limited to specific few types of supervisions and do not fully align with the goal of pre-training task-agnostic medical image representations. Borrowing the insight from the language domain (Brown et al., 2020), since the downstream tasks are unknown during the pre-training stage, the model needs to *encounter*

---

[1]Models and codes will be released upon acceptance.

*as many diverse types of data and annotations as possible*, rather than being restricted to a limited set, to to acquire as many abilities as possible in the pre-training stage.

In this work, we take the first step toward exploring a new objective: *unifying all common types of supervisions in the medical image domain through a same scalable way in one model*. This goal is quite challenging, as it requires the framework to be both *unified* and *effective*. Firstly, the framework needs to adopt a cohesive approach rather than implementing complex designs tailored to the characteristics of the data and annotations. Real-world medical data is highly diverse, and new data types may emerge; a unified framework allows the model to encounter more data types and offers better scalability. Additionally, the framework must be effective, ensuring that various heterogeneous data and supervisions, with distinct characteristics, can synergistically assist unknown downstream tasks, which is the essence of pre-training models. This goal in the general vision domain also remains challenging and unsolved (Bai et al., 2024; Wang et al., 2023). Here, we focus on medical images, as medical datasets often have diverse annotations and small individual sizes, necessitating such a unified framework. Furthermore, we concentrate on 2D X-rays, given the relatively good public availability of this medical modality, and because 2D data provides a cleaner setting to study the synergistic effects of different supervisions within a unified framework. Besides, X-rays are a common diagnostic modality, making the pre-training techniques for X-rays clinically significant.

We propose a **U**nified **m**edical **i**mage pre-training framework, *UmiF*, aiming at tackling all common types of supervisions, including (1) image and patch-level self-supervisions; (2) external supervisions such as paired reports, captions, segmentation annotations, and classification labels. The design principle behind *UmiF* is simple: once converted into token embeddings in a unified space, all supervisions can be effectively utilized via contrastive learning and mask modeling with a same way, making them collectively contributing to the development of a robust medical image representations. In *UmiF*, an image and its supervision, such as the segmentation annotation, form an input pair. This pair is then tokenized separately and concatenated into a sequence of input tokens. *UmiF* introduce a novel flexible token grouping strategy to randomly split input tokens into two groups. These groups are used as a positive pair for contrastive learning, and two incomplete views for mask modeling. Besides, all tokens are processed by a single backbone, enabling effective fusion of all signals.

*UmiF* well addresses the above two requirements. Specifically, although the data types across various datasets are highly diverse, we abstract three modalities (i.e., radiology, language and segmentation mask) and introduced modality-specific tokenizers. This design avoids excessive data-specific operations and facilitates the transformation of data into a unified token space, providing the basis for the cohesive modeling for all data types. Besides, the random token grouping strategy makes the data views in contrastive learning and mask modeling highly flexible and varied, thereby effectively covering a wide range of supervisions and largely enriching learning tasks. This enables thorough exploration of the data, enhancing the effectiveness of *UmiF*. Our contributions are summarized as:

- We introduce a novel pre-training framework *UmiF* that can unify all common types of supervisions in the medical image domain through a same way and one model. *UmiF* introduces a unified token space and a novel flexible token grouping strategy, making the framework unified and effective at the same time.

- To fully exploit the advantages of *UmiF*, we collect a large-scale pre-training datasets based on public datasets, comprising 1.66M pairs (include 1M images), significantly surpassing previous efforts, which mostly limited to 380K image-report pairs or 838K images.

- By overcoming several challenges when implementing *UmiF*, we obtain and release a generalized pre-trained encoder for medical images based on Vision Transformer (ViT). Even when compared to previous methods with many data- and supervision-specific designs, *UmiF* reaches SOTA performances on most of downstream tasks, including classification, semantic segmentation, object detection, retrieval and VQA, showing outstanding capabilities in both image and patch level.

## 2  PRE-TRAINING DATASETS AND PROCESSING

Regarding pre-training data, we focus on 4 types of input pairs: image-report, image-caption, image-class and image-segment. Each type includes multiple public datasets, and we provide a processing pipeline that uniformly converts different datasets into *UmiF* inputs. Previous pre-training works

based on public data were mostly limited to MIMIC-CXR dataset (Johnson et al., 2019a) with 220K image-report pairs or PadChest dataset (Bustos et al., 2019) with 160K pairs, or image-only pre-training (Pérez-García et al., 2024) with 838K images. Our data used in pre-training comprises 1.66M pairs (include 1M images), significantly surpassing previous efforts. Included datasets used in our pre-training framework *UmiF* are listed in Table 1. And more details are in Appendix A.

**Image-Report** We include two large real-world chest X-ray (CXR) image-report datasets, i.e., MIMIC-CXR with English reports and PadChest with Spanish reports. For Spanish reports in PadChest, we translate them into English with GPT-4, and further ask GPT-4 to polish the translated English reports, resulting in two versions of reports. For other datasets, they originally only have class labels and we ask GPT-4 to generate reports consistent with labels.

**Image-Caption** These data mainly come from figures and captions in biomedical papers and we only use the radiology images provided by the datasets. Comparing with image-report datasets, which contain CXR and detailed findings from doctors, image-caption data contain different types of images in papers and simpler descriptions.

**Image-Class** The classes in these data are about disease types. Different datasets may use varying names for the same disease, so we standardized the labels across these ten datasets.

**Image-Segment** These two datasets contain CXR images accompanied by detailed annotations regarding the locations of pathologies, represented via their coordinates. Similar to image-class datasets, pathology types are also standardized across datasets.

Table 1: Statistics on the datasets used in pre-training in *UmiF*.

| Input pair type | Datasets | # Sample |
|---|---|---|
| Image-Report | Brax Reis et al. (2022), Candidptx Feng et al. (2021), Chexpert Irvin et al. (2019), Jfhealthcare Healthcare (2020), Nih Wang et al. (2017), Vindr Nguyen et al. (2020), Padchest Bustos et al. (2020), Mimic Johnson et al. (2019b) | 668K |
| Image-Caption | ROCO Pelka et al. (2018), MedICaT Subramanian et al. (2020) | 229K |
| Image-Class | Brax Reis et al. (2022), Candidptx Feng et al. (2021), Chexpert Irvin et al. (2019), Jfhealthcare Healthcare (2020), Midrc Tsai et al. (2021), Mimic Johnson et al. (2019b), Mura Rajpurkar et al. (2017), Nih Wang et al. (2017), Padchest Bustos et al. (2020), Vindr Nguyen et al. (2020) | 761K |
| Image-Segment | CheXlocalize Saporta et al. (2022b), ChestX-ray14 Wang et al. (2017) | 2K |

## 3 UNIFY ALL COMMON SUPERVISIONS FOR MEDICAL IMAGE REPRESENTATION LEARNING

In this section, we present the methodology of *UmiF*, as illustrated in Figure 1. We first show how different types of input pairs are converted into tokens in Section 3.1, providing basis to utilize multiple supervision for training in the same way. Then, we introduce the novel flexible grouping strategy in Section 3.2, which is the key to realize learning tasks and interactions among input signals. Finally, we show the enabled learning tasks and architectures used by *UmiF* in Section 3.3.

### 3.1 UNIFIED TOKEN SPACE FOR VARIOUS INPUT PAIRS

To integrate diverse types of supervisions into a unified framework, we adopt an idea similar to some previous works (Wang et al., 2022b; Zhang et al., 2023) in the general domain, by converting all input data into token embeddings. Differently, they focus on generation tasks, while we aim to learning a medical image encoder with multi-level capabilities.Our overall approach involves first designing a modality abstraction, mapping all input data listed in Table 1 to three different modalities (radiology, language, and segmentation masks). Then, for each modality, we introduce specific tokenizers to convert them into token embeddings.

**View Input Pairs as Three Modalities** All images in inputs are radiology image, belonging to the radiology modality. Supervisions in image-report and image-caption datasets (i.e., reports and

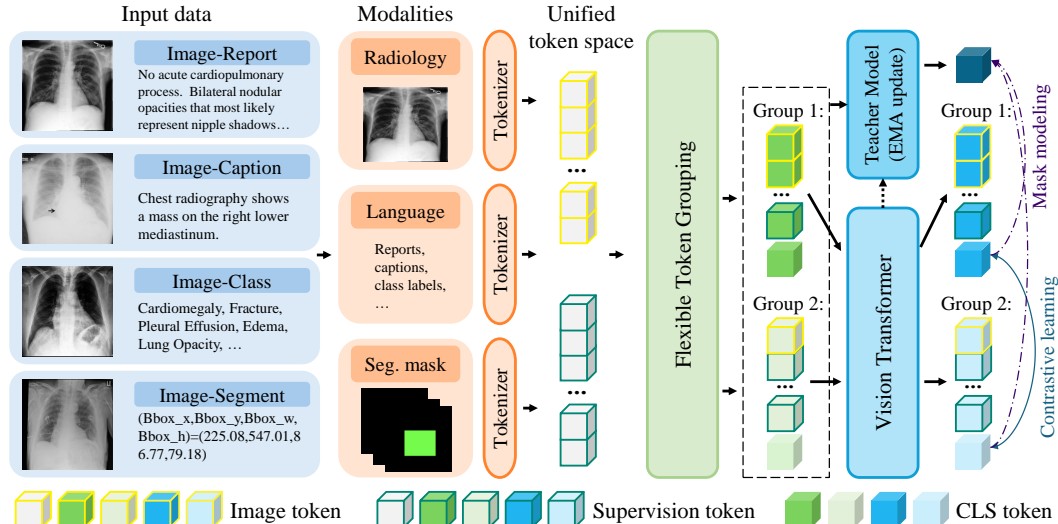

Figure 1: Illustration for the proposed *UmiF* framework, a unified framework for all common types of supervisions in the medical image domain. *UmiF* covers various datstes with 4 different input pair types, comprising 1.66M pairs. By abstracting 3 kinds of modalities and using modality-specific tokenizers, all data can by unified in a token space. In *UmiF*, an image and its supervision form an input pair. This pair is then tokenized separately and concatenated into a sequence of input tokens. Then, *UmiF* employs a novel flexible token grouping strategy to randomly split input tokens into two groups, serving as a positive pair for contrastive learning, and two incomplete views for mask modeling. This strategy, together with the unified token space, flexibly enables and enriches various learning tasks, making the model to fully exploit the information in the pre-training data, thereby allowing diverse datasets to synergistically contribute to learning one transformer model with robust medical image representation capabilities.

captions) are texts, belonging to the language modality. For class labels in image-class datasets, we employ fixed templates (see Appendix A.3) to convert existing disease labels into a text passage, thus categorizing them as the language modality. For image-segment data, to better integrate pixel-level supervision, we do not simply convert coordinates into text. Instead, we generate a segmentation mask with the same size as the image based on the coordinates. As shown in the bottom left corner of Figure 1, the mask has a black background, and each abnormality is marked with a different color patch. Since the mask is an RGB image, it belongs to the third modality, i.e., segmentation mask.

**Modality-Specific Tokenizers** For the language modality, we use tokenizers from BioClinical-BERT (Alsentzer et al., 2019), including a segmentation unit to convert texts into segments, and a word embedding layer to map segments into token embeddings. For the radiology and segmentation mask modalities, since both are images, we apply the same patching operation and tokenizer from DeiT (Touvron et al., 2021) to convert input patches into token embeddings. Separate tokenizers are used for radiology and segmentation mask, but they use the same initialization. In this way, data of three modalities can be converted into a sequence of token embeddings, with each embedding being a one-dimensional vector of dimension $D$ (denoted as $\mathbf{x} \in \mathcal{R}^D$), and the number of tokens may vary for each modality.

Through these two steps, we convert diverse input pairs into token embeddings of the same dimension, allowing these tokens to be processed by a unified transformer model. Besides, by mapping different data into a unified token space, we can conveniently design token-based learning tasks without focusing on the specifics of each data type and modality, facilitating a multimodal learning approach that addresses the varied nature of the data in radiology image and annotations.

## 3.2 FLEXIBLE TOKEN GROUPING STRATEGY ENRICHES DIFFERENT LEARNING TASKS

To accommodate different learning tasks with a unified approach, we designed a flexible grouping strategy that divides tokens into two groups for contrastive learning and mask modeling. Specifically,

a input sample to *UmiF* is image-supervision pair, with their token embeddings denoted as $\mathbf{X}^i \in \mathcal{R}^{n^i \times D}$ and $\mathbf{X}^s \in \mathcal{R}^{n^s \times D}$, where $n^i$ and $n^s$ are the number of image tokens and supervision tokens, respectively. Then, we introduce a set of randomly sampled binary bits $\mathbf{b} = \mathrm{Concat}(\mathbf{b}^i, \mathbf{b}^s)$ where $\mathbf{b}^i \in \{0,1\}^{n^i}$ and $\mathbf{b}^s \in \{0,1\}^{n^s}$. According to whether the binary bit at each corresponding position in $\mathbf{b}$ is 0 or 1, we can divide the tokens into two groups. We use $\mathbf{X1} = \mathrm{Concat}(\mathbf{X1}^i, \mathbf{X1}^s)$ to denote token embeddings in group 1, which is a concatenation of tokens embeddings from $\mathbf{X}^i$ and $\mathbf{X}^i$ at positions where the corresponding binary bit is 1. Similarly, token embeddings in group 0 are denoted as $\mathbf{X0} = \mathrm{Concat}(\mathbf{X0}^i, \mathbf{X0}^s)$. Therefore, tokens are split into two groups according to $\mathbf{b}$.

Here, we use $r$ to represent the ratio of 1 in $\mathbf{b}^i$ and let the ratio of 1 in $\mathbf{b}^s$ be $1 - r$, and then sampling $\mathbf{b}$ according to $r$. Setting $r = 1$ means image and supervision tokens are separated into two groups. Since these two groups are used as the positive pair in contrastive learning, when $r = 1$ and the supervision is language, *UmiF* degrades to vision-language (VL) learning in CLIP. Beyond this point, other values of $r$ produce the mixed view composed of partial image and supervision tokens (as shown in Figure 1). This interesting design allows more diverse views and enriches the learning tasks with many possibilities, surpassing previous VL learning approaches. To ensure more cross-modality information can be leveraged by *UmiF*, we employ the following method to set the ratio $r$. With a certain probability, $r$ is set to 1 and in remaining cases, $r$ is randomly sampled from $[0, 1]$. We provide detailed ablations on the probability in experiments in Section 4.4.

### 3.3 MODEL ARCHITECTURE AND LEARNING TASKS

Following (Wang et al., 2022d; Zhang et al., 2023), all token embeddings are processed by one model for better information fusion. *UmiF* uses ViT (Dosovitskiy et al., 2020a) as it is proven to be effective for pre-training with large-scale data in many previous works (Wang et al., 2022c; Zhang et al., 2023). After pre-training, we freeze the weights of the model and use it for different downstream tasks. To obtain global information, we use two CLS tokens for $\mathbf{X1}$ and $\mathbf{X0}$, separately. They are encoded by the ViT model and denoted as $\mathbf{f1}$ and $\mathbf{f0}$.

**Sampling Pairs and Constructing Batch** Since a large number of diverse datasets are incorporated by *UmiF*, batch data sampling and construction strategy is critical for the final performance. The challenges include balancing among data types and datasets to avoid overfitting on dominant ones. We choose to first sample input pair types, where the type with larger dataset size has higher probability to be sampled, and then sample pairs with the corresponding type (see Algorithm 1 in Appendix for details). Therefore, all samples have approximately the same probability of being selected, ensuring coverage, while allowing for a variety of data pair types within a single batch. Let $B$ denote the number of sampled images, together with their supervisions, $2B$ data points in total are obtained in a training minibatch. Next, we introduce learning objectives in *UmiF*.

**Contrastive Learning** Contrastive learning learns representations by maximizing agreement between positive pairs via a contrastive loss in the latent space. Here, the positive pair is constructed via the flexible token grouping strategy explained in the last sub-section, and remaining $2(B-1)$ data points within the minibatch are considered as negative examples. We them apply an alignment loss for contrastive learning, borrowed from SimCLR (Chen et al., 2020). Specifically, features of one positive pair is represented by $(\mathbf{f1}_i, \mathbf{f0}_j)$ with $i, j$ being the index of data point, and sim is the cosine similarity. The loss function is

$$\mathcal{L}_{align} = \frac{1}{B} \sum_{\text{B positive pairs}} \ell(\mathbf{f1}_i, \mathbf{f0}_j) + \ell(\mathbf{f0}_j, \mathbf{f1}_i),$$

$$\text{where } \ell(\mathbf{f}_i, \mathbf{f}_j) = -\log \frac{\exp(\mathrm{sim}(\mathbf{f}_i, \mathbf{f}_j)/\tau)}{\sum_{k=1}^{2B} \mathbb{1}_{[k \neq i]} \exp(\mathrm{sim}(\mathbf{f}_i, \mathbf{f}_k)/\tau)}. \tag{1}$$

$\tau$ is a temperature parameter that scales the distribution of distances, and $\mathbb{1}_{[k \neq i]}$ is an indicator function that equals 1 when $k \neq i$ and 0 otherwise. This formulation encourages representations of positive pairs to be more similar to each other than to any other example in the minibatch, effectively learning from the structure inherent within the data itself. Note that by using this method to construct positive and negative pairs, images in the same batch can also be the negative view. Therefore, image-level self-supervision is also included in *UmiF*. Also, since a batch of data may come from different dataset, *UmiF* also enables learning signals cross multiple datasets.

**Mask Modeling** Another self-supervision task *UmiF* considered is mask modeling. Specifically, the binary bits $\mathbf{b}$ can also be regarded as masks and we consider the consistency between the complete view and the masked view. The objective is to ensure that the model can effectively learn invariant or robust features that are representative of the underlying content, despite variations in visibility due to masking. Specifically, during the training process, we maintain a student network, and a teacher network, which is updated by the exponential moving averaged (EMA) over the student network. Different from the masked inputs to the student model, the input token embeddings of the teacher model are directly concatenation of $\mathbf{X}^i$ and $\mathbf{X}^s$. Finally, the teacher model outputs a CLS token $\mathbf{t}$. We then apply the consistency loss from BYOL (Grill et al., 2020):

$$\mathcal{L}_{con} = \frac{1}{B} \sum_{i=1}^{B} 2 - 2 \frac{\mathbf{t}_i^\top \mathbf{f0}_i}{||\mathbf{t}_i|| \cdot ||\mathbf{f0}_i||} + 2 - 2 \frac{\mathbf{t}_i^\top \mathbf{f1}_i}{||\mathbf{t}_i|| \cdot ||\mathbf{f1}_i||}. \tag{2}$$

In this way, the student network is required to predict the teacher network representation of the same input under a complete view.

**Unify Various Supervisions and Learning Tasks** We now explain why *UmiF* can enable a variety of different learning tasks and unify supervisions through these learning tasks. As mentioned before, vanilla VL and beyond are incorporated by *UmiF*, and images in the same batch can serve as negative views, accounting for the image-level self-supervision. The combination of flexible batch sampling methods and token grouping strategy provides a multiplicative increase in learning task diversity. Besides, inputs also contains image-segmentation pairs. Here, when and image and its supervision are in separate groups, mask modeling force the model to predict the segmentation annotation. When they are mixed, the model needs to reconstruct the image. Thus, both patch-level self-supervision and external supervision are inherently included in *UmiF*. Thus, *UmiF* is highly flexible covering various forms of supervision, which makes it ideally suits for medical image pre-training, where datasets often have diverse annotations and small individual sizes. Additionally, the design of pre-training tasks endows the model with both image and patch-level capabilities, enabling it to handle diverse downstream tasks in medical image analysis.

## 4 EXPERIMENTS

### 4.1 PRE-TRAINING CONFIGURATION

In the pre-training stage, we employ ViT (Dosovitskiy et al., 2020b) as our backbone. Our *UmiF* model is obtained after 50 epochs training on 16 V100 GPUs with a batch size of 128 per GPU using *UmiF*. We utilize AdamW (Loshchilov & Hutter, 2017) as the optimizer, setting the learning rate to $4e^{-5}$ and the weight decay to $5e^{-2}$. A linear warm-up and cosine annealing scheduler are also deployed in this process.

### 4.2 DOWNSTREAM TASKS

**Medical Image Linear Classification** We conduct medical image linear classification on three representative dataset: CheXpert (Irvin et al., 2019), RSNA (Shih et al., 2019), and COVIDx (Wang et al., 2020). We adopt data split strategies in (Huang et al., 2021; Zhang et al., 2020; Wang et al., 2022a) for the datasets. Meanwhile, we keep the pre-trained ViT vision encoder fixed and solely training a linear classification head initialized randomly for the classification task with varying amounts of training data on each dataset. We report the AUC scores (AUC) on CheXpert and RSNA and accuracy (ACC) on COVIDx as the evaluation metric following (Huang et al., 2021; Wang et al., 2022a).

**Medical Image Semantic Segmentation** Following (Wang et al., 2022a; Huang et al., 2021), we conduct medical image semantic segmentation on RSNA (Shih et al., 2019) and the SIIM (Steven G. Langer & George Shih, 2019) datasets. We keep the pre-trained vison backbone frozen and only update the decoders of U-Net during the fine-tuning. The segmentation performance is evaluated using Dice scores (Dice).

**Medical Image Object Detection** Following (Wang et al., 2022a), we conduct medical image object detection on RSNA (Shih et al., 2019). We utilize YOLOv3 (Redmon & Farhadi, 2018) as the detection architecture, using our pre-trained vision encoder as the backbone and only updating the

Table 2: Linear classification results for CheXpert, RSNA, and COVIDx datasets with 1%, 10%, and 100% training data. The best results are highlighted in bold.

| Method | CheXpert (AUC) | | | RSNA (AUC) | | | COVIDx (ACC) | | |
|---|---|---|---|---|---|---|---|---|---|
| | 1% | 10% | 100% | 1% | 10% | 100% | 1% | 10% | 100% |
| Random Init | 56.1 | 62.6 | 65.7 | 58.9 | 69.4 | 74.1 | 50.5 | 60.3 | 70.0 |
| ImageNet Init | 74.4 | 79.7 | 81.4 | 74.9 | 74.5 | 76.3 | 64.8 | 78.8 | 86.3 |
| *CNN-based* | | | | | | | | | |
| GLoRIA (Huang et al., 2021) | 86.6 | 87.8 | 88.1 | 86.1 | 88.0 | 88.6 | 67.3 | 77.8 | 89.0 |
| ConVIRT (Zhang et al., 2020) | 85.9 | 86.8 | 87.3 | 77.4 | 80.1 | 81.3 | 72.5 | 82.5 | 92.0 |
| GLoRIA-MIMIC (Huang et al., 2021) | 87.1 | 88.7 | 88.0 | 87.0 | 89.4 | 90.2 | 66.5 | 80.5 | 88.8 |
| MedKLIP (Wu et al., 2023) | 86.2 | 86.5 | 87.7 | 87.3 | 88.0 | 89.3 | 74.5 | 85.2 | 90.3 |
| MGCA (Wang et al., 2022a) | 87.6 | 88.0 | 88.2 | 88.6 | 89.1 | 89.9 | 72.0 | 83.5 | 90.5 |
| Med-UniC (Wan et al., 2024) (ResNet-50) | 88.2 | 89.2 | 89.5 | 89.1 | 90.4 | 90.8 | 76.5 | 89.0 | 92.8 |
| *ViT-based* | | | | | | | | | |
| MRM (Zhou et al., 2023a) | 88.5 | 88.5 | 88.7 | 91.3 | 92.7 | 93.3 | 66.9 | 79.3 | 90.8 |
| MGCA (ViT-B/16) (Wang et al., 2022a) | 88.8 | 89.1 | 89.7 | 89.1 | 89.9 | 90.8 | 74.8 | 84.8 | 92.3 |
| Med-UniC (Wan et al., 2024) (ViT-B/16) | **89.4** | 89.7 | 90.8 | 91.9 | 93.1 | 93.7 | **80.3** | **89.5** | 94.5 |
| *UmiF* (ViT-B/16) | 89.1 | **90.1** | **90.9** | **92.2** | **93.4** | **93.8** | 79.6 | 88.6 | **94.8** |

detection head during fine-tuning. Mean Average Precision (mAP) with IOU thresholds 0.4∼0.75, is adopted to evaluate the detection task.

**Medical Image Zero-shot Classification**   Following (Huang et al., 2021; Wan et al., 2024),We conduct this experiment on the CXP500 (Saporta et al., 2022a), which is the test set of CheXlocalize. It includes 500+ CXR images with clinician annotated disease label. The results are represented as the macro average of AUC across all categories.

**Medical Visual Question Answer**   We conduct Medical Viusal Question Answer on VQA-RAD (Lau et al., 2018). VQA-RAD has 315 radiology images with 3064 question-answer pairs, with 451 pairs used for testing. There are two types of questions: closed-ended questions that have limited answer choices (e.g. "yes" or "no") and open-ended questions that VQA models are required to generate answers in free text, which are more challenging. Following (Li et al., 2023; Chen et al., 2022), we add a decoder and fintune the whole model.

For Medical Image Linear Classification, Semantic Segmentation and Object Detection, we fine-tune with $1\%, 10\%, 100\%$ of the training data.

### 4.3   COMPARISON TO PREVIOUS STATE-OF-THE-ART

**Medical Image Linear Classification**   To evaluate the effectiveness of the visual representations learned by the *UmiF*, we conduct linear classification tasks on three medical datasets: CheXpert (Irvin et al., 2019), RSNA (Shih et al., 2019), and COVIDx (Wang et al., 2020). As demonstrated in Tab 2, our *UmiF* model exhibits best performance in most settings. It is worth noting that the some baselines use designs tailored to specific data and annotations. Med-UniC belongs to a multi-stage pre-training paradigm and focuses on unifying cross-lingual text (English and Spanish), so they basically employ back-translation as an augmentation. MedKLIP and MGCA utilize VL pre-training with disease-level annotations. In contrast, we target on a unified framework for incorporating as many diverse data types as possible, so such specific designs are not utilized by us. These designs are largely orthogonal with our method, but they are not consistant with the focus of this work about studying a unified framework to make data synergistically benefit downstream tasks. Even without these specific operations, *UmiF* shows very competitive performance, well demonstrating that representations encoded by *UmiF* is discriminative in terms of disease and abnormality types, and *UmiF* is able to learn robust and task-agnostic representations for medical images.

**Medical Image Semantic Segmentation and Object Detection**   We extend our evaluation of *UmiF*'s representations to include segmentation and detection tasks in Tab 3. Remarkably, *UmiF* surpasses all SOTA methods in every evaluated data subset for all dataset. Notably, for segmentation tasks, our method outperforms Med-UniC with ViT-B/16 backbone with +1.4%, +0.3%, +0.9% Dice on SIIM dataset, +0.8%, +0.5%, +0.3% Dice on RSNA dataset under the 1%, 10%, 100%

Table 3: Results of semantic segmentation (Dice) on SIIM and RSNA datasets and object detection (mAP) on RSNA dataset. The best results for each setting are highlighted in bold.

| | Semantic Segmentation | | | | | | Object Detection | | |
| | SIIM | | | RSNA | | | RSNA | | |
| Method | 1% | 10% | 100% | 1% | 10% | 100% | 1% | 10% | 100% |
|---|---|---|---|---|---|---|---|---|---|
| Random | 9.0 | 28.6 | 54.3 | 6.9 | 10.6 | 18.5 | 1.0 | 4.0 | 8.9 |
| ImageNet | 10.2 | 35.5 | 63.5 | 34.8 | 39.9 | 64.0 | 3.6 | 8.0 | 15.7 |
| ConVIRT (Zhang et al., 2020) | 25.0 | 43.2 | 59.9 | 55.0 | 67.4 | 67.5 | 8.2 | 15.6 | 17.9 |
| GLoRA (Huang et al., 2021) | 35.8 | 46.9 | 63.4 | 59.3 | 67.5 | 67.8 | 9.8 | 14.8 | 18.8 |
| GLoRIA-MIMIC (Huang et al., 2021) | 37.4 | 57.1 | 64.0 | 60.3 | 68.7 | 68.3 | 11.6 | 16.1 | 24.8 |
| MGCA (Wang et al., 2022a) | 49.7 | 59.3 | 64.2 | 63.0 | 68.3 | 69.8 | 12.9 | 16.8 | 24.9 |
| MedKLIP (Wu et al., 2023) | 50.2 | 60.8 | 63.9 | 66.2 | 69.4 | 71.9 | 8.9 | 16.3 | 24.5 |
| Med-UniC (Wan et al., 2024) (ResNet-50) | 56.7 | 62.2 | 64.4 | 72.6 | 74.4 | 76.7 | 16.6 | 22.3 | 31.1 |
| Med-UniC (Wan et al., 2024) (ViT-B) | 62.1 | 67.3 | 71.5 | 75.6 | 76.6 | 77.9 | - | - | - |
| *UmiF* (ViT-B) | **63.5** | **67.6** | **72.4** | **76.4** | **77.1** | **78.2** | **18.7** | **23.4** | **32.2** |

training ratio respectively. Meanwhile, for detection tasks, *UmiF* also achieves +2.1%, +1.1%, +1.1% performance gain over the previous method. The significant improvement demonstrate *UmiF* has much better patch-level capacity comparing with previous VL-based models. This result indicate the importance of incorporating segmentation annotations in pre-training, the efficiency and effectiveness of *UmiF* in utilizing supervisions on segmentation.

**Medical Image Zero-shot Classification**    To assess the efficacy of the visual-textual representation capabilities of *UmiF*, we executed a zero-shot image classification task using the CXP500 dataset. The zero-shot learning paradigm is particularly challenging, as it requires the model to correctly classify images it has never seen during training, which is a testament to the generalizability of the learned representations. As detailed in Table 4, *UmiF* not only meets but exceeds the performance of all current SOTA methods when evaluated on the CXP500 dataset. This superior performance is indicative of *UmiF*'s robust understanding of visual and textual data, capturing nuanced relationships between the two modalities without the need for explicit example-based learning for each class.

**Medical VQA** Consistent with previous research (Chen et al., 2022; Li et al., 2023), we adopt accuracy as the performance metric. We treated VQA as a generative task by calculating similarities between the generated answers and candidate list answers, selecting the highest score as the final answer. As illustrated in Tab 5, *UmiF* outperforms all other methods on VQA-RAD, and yields the best accuracy for open-ended and closed-ended answers. *UmiF* achieves an absolute margin of 0.8% in Open-ended, 0.7% in Closed-ended, 0.7% Overall over the SOTA method, MUMC. These results suggest that representations encoded by *UmiF* have rich semantic information, verifying that *UmiF* can improve medical image understanding over previous pre-training methods.

## 4.4 FURTHER ANALYSIS

**The Probability in Flexible Token Grouping Strategy** As illustrated in Section 3.2, *UmiF* apply a certain probability to let $r$ set to 1, so *UmiF* degrades to VL learning in CLIP, ensuring *UmiF* can make use of cross-modality information. Table 6 demonstrates the influence of probability in *UmiF* on RSNA classification (1%). We see that when setting the probability to 0.2, *UmiF* achieves the best performance in RSNA 1% classification. Note that although we do not include results when the probability is greater than 0.8 in the table, we observe large performance descent for those cases. Overall, these results indicate the importance of introducing the flexible grouping strategy to enable sampling different kinds of positive pairs.

**Ablation Study on Unifying Supervisions** In order to show the significance in unifying various supervisions in pre-training and investigate whether *UmiF* can uncover their synergistic effects, we conduct ablation studies, where only one type of input pairs in included in supervision. As illustrated in Tab 7, we compare these settings in three different downstream tasks (RSNA linear classification, RSNA semantic segmentation and VQA-RAD VQA). Overall, we observe that model trained with

Table 4: Results of Zero-shot Classification on CXP500. The best results for each setting are highlighted in bold.

| Method | CXP500 AUC |
|---|---|
| MGCA$^\star$ (Wang et al., 2022a) | 72.1 |
| MedKILP$^\star$ (Wu et al., 2023) | 70.5 |
| MRM (Zhou et al., 2023a) | 65.2 |
| Med-UniC (Wan et al., 2024) | 75.4 |
| *UmiF* | **76.5** |

Table 5: Results of VQA on VQA-RAD. The best results for each setting are highlighted in bold.

| Method | VQA-RAD | | |
|---|---|---|---|
| | Open | Closed | Overall |
| CPRD (Liu et al., 2021) | 61.1 | 80.4 | 72.7 |
| PubMedCLIP (Eslami et al., 2021) | 60.1 | 80.0 | 72.1 |
| MTL (Cong et al., 2022) | 69.8 | 79.8 | 75.8 |
| M3AE (Chen et al., 2022) | 67.2 | 83.5 | 77.0 |
| MUMC (Li et al., 2023) | 71.5 | 84.2 | 79.2 |
| *UmiF* | **72.3** | **84.9** | **79.9** |

Table 6: The influence of probability in Flexible Token Grouping Strategy on RSNA classification (1%). The top-2 results for each setting are highlighted in bold

| | 0.0 | 0.1 | 0.2 | 0.3 | 0.4 | 0.5 | 0.6 | 0.7 | 0.8 |
|---|---|---|---|---|---|---|---|---|---|
| RSNA (1%) AUC | 91.5 | 91.2 | **92.2** | 91.4 | 90.8 | 90.5 | 91.3 | **92.0** | 90.9 |

all kinds of input pairs achieves the best performance on all tasks. This verifies our motivation that for building task-agnostic medical image representations, the model needs to see as many diverse data as possible during pre-training, indicating the importance of unifying all kinds of supervisions in medical image pre-training. Results also demonstrate the effectiveness of our *UmiF* on utilizing those supervisions. *UmiF* can make data with distinct character synergistically contribute to multiple downstream tasks.

Besides, we also observe that different data contributes to different downstream tasks. Specifically, image-report and image-class data has significant impact on linear classification tasks. Meanwhile, image-caption contributes largely to VQA tasks. These results suggest that when datasets in pre-training is more closer to downstream tasks, more benefits are gained via pre-training. However, downstream tasks is often unknown during pre-training and medical image analysis tasks are highly diverse, thus requiring pre-training frameworks to include more and more kinds of data. This underscores the importance of the direction explored by our work.

## 5 RELATED WORK

**Medical Vision-Language Pre-training** The intricate nature of medical reports, coupled with the scarcity of extensive medical image-text datasets, has constrained research in the field of medical Vision-and-Language Pretraining (VLP). ConVIRT (Zhang et al., 2020) learns medical visual repre-

Table 7: Ablation study on unifying supervisions. The row of the input pair type contains results of the model pre-trained only with that type of data.

| Input pair type | RSNA (AUC) | | | RSNA (Dice) | | | VQA-RAD (ACC) |
|---|---|---|---|---|---|---|---|
| | 1% | 10% | 100% | 1% | 10% | 100% | Overall |
| Image-Report | 91.0 | 92.6 | 93.1 | 75.9 | 76.5 | 77.4 | 76.2 |
| Image-Caption | 86.5 | 87.3 | 88.4 | 69.5 | 70.4 | 70.8 | 77.3 |
| Image-Class | 90.5 | 91.7 | 92.4 | 73.2 | 74.6 | 75.1 | 74.1 |
| Image-Segment | 85.1 | 85.3 | 86.0 | 70.2 | 71.5 | 71.8 | 68.3 |
| **All** | **92.2** | **93.4** | **93.8** | **76.4** | **77.1** | **78.2** | **79.9** |

sentations by exploiting naturally occurring paired descriptive text. Based on this, Gloria (Huang et al., 2021) learns global and local representations by contrasting image sub-regions and words in the paired report. MGCA (Wang et al., 2022a) further exploits the high-level semantic correspondences between inter-subject relationships, such as those related to disease. MedKLIIP (Wu et al., 2023) involves the extraction of entities pertinent to the medical field. Meanwhile, MRM (Yang et al., 2023), substituted the alignment task with one focused on reconstruction, which involved handling masked tokens within both visual and textual modalities. Med-UniC (Wan et al., 2024) integrates multimodal medical data from the two most prevalent languages, English and Spanish. However, the availability of publicly accessible medical imaging report datasets has restricted the progress of visual representation learning techniques. Exploring how to utilize diverse annotated data remains an issue that needs to be addressed.

**Self-Supervised Learning in Medical imaging** Recent work in self-supervised learning is using discriminative signals between images or groups of images to learn features (Chen et al., 2020; He et al., 2020; Grill et al., 2020). In medical domain, self-supervised learning has also achieved numerous successes. (Chaitanya et al., 2020) develops a novel method to enhance the contrastive learning framework tailored for the task of segmenting three-dimensional medical images. MICLe (Azizi et al., 2021) leverages the availability of multiple images depicting the underlying pathology from each patient case, which constructs more informative positive pairs for self-supervised learning. Swin UNETR (Tang et al., 2022) introduce a novel self-supervised learning framework with tailored proxy tasks for medical image analysis. Self-supervised learning has made significant contributions to the field of medical image processing by reducing the reliance on labeled data, meanwhile enhancing feature representation learning.

**Unified Frameworks** In the field of Natural Language Processing (NLP), recent research has been moving towards unifying a range of tasks from natural language understanding to generation into a text-to-text framework, or treating them as language modeling challenges. Building upon this concept, (Cho et al., 2021; Yang et al., 2021) have introduced multimodal pretraining models that are based on text generation. Furthermore, (Jaegle et al., 2021;?) have developed a straightforward framework capable of handling inputs from multiple modalities through a consistent representation in byte sequences. OFA (Wang et al., 2022b) unifies a diverse set of crossmodal and unimodal tasks, including image generation, visual grounding, image captioning, image classification, language modeling, etc., in a simple sequence-to-sequence learning framework. Painter (Wang et al., 2023) is a generalist model which addresses these obstacles with an "image"-centric solution, which redefines the output of core vision tasks as images, and specify task prompts as also images. (Huang et al., 2024; Peng et al., 2023) introduce a Multimodal Large Language Model (MLLM) that can perceive general modalities. Recently, (Yi et al., 2023; Chen et al., 2023) further explore the possibility of leveraging pre-trained VLMs as medical foundation models for building general purpose medical AI. Given the variety of annotated data available for medical images, it is essential to fully leverage these resources to construct a unified model.

# 6 CONCLUSION, LIMITATION AND FUTURE WORK

In this paper, we introduce an Unified medical image pre-training framework, namely *UmiF*, assembling all common type of supervision for medical images in a same scalable way. By converting all signals into token embeddings and leveraging a novel flexible grouping strategy, *UmiF* successfully integrates self-supervisions like masking and recovering, as well as external supervisions, including reports, captions, class labels and segmentation annotations. The pre-trained encoder *UmiF* reaches SOTA performance on various downstream tasks, well demonstrating the importance of unifying various signals and supervisions in one framework and the effectiveness of the *UmiF* framework in uncovering synergistic effects of distinct pre-training datasets to multiple downstream tasks.

**Limitation and Future Work** One limitation is that our model is only trained on public datasets, which might exist region bias, since radiology device and experts in underdeveloped areas are insufficient, and data collection process in these areas is almost infeasible. This limitation can be addressed by including more private data, which one of our future work. Besides, developing AI models in radiology, which is the research focus in this paper, offers large potential in solving this radiology resource shortage and imbalance issue.

**Ethics Statement** The datasets utilized in this paper are public datasets, making the results reproducible by the boarder research community. Medical image analysis model might have negative societal impacts, such as provide incorrect diagnosis. This can be mitigated by using medical image models in a careful way, where human doctor control is always available and decisions made by the model cannot directly effect treatments to patients.

**Reproducibility Statement** To ensure reproducibility, we have provided details about *UmiF* in the main paper and appendix, including detailed designs, datasets, experiment setting, prompts when using LLMs, et al. Furthermore, we plan to release all our code and model checkpoints upon the acceptance.

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

ROADMAP OF APPENDIX

The structure of the appendix is delineated as follows: Descriptions of the used dataset details are provided in the Section A.

# A    PRE-TRAINING DATASETS

## A.1    IMAGE-REPORT DATASET

We divide our Image-Report Dataset to three group: CXR images with original English reports 2, CXR images with original Spanish reports 3 and CXR images with generated reports 4. We use prompts as follows to generate reports and translations:

'You are a senior radiologist proficient in Spanish and English, specializing in interpreting Chest X-rays. Here is a section of a Spanish report: <Spanish> . . siluet cardi mediastin dentr normal . . cambi pulmonar cronic . . . sen costofren libr . . . no sign enfermed metastas. </Spanish> Please provide the English translation in xml format in English tag: <English></English> And then polish the language of the report as a native radiologist in Report tag: <Report></Report>'

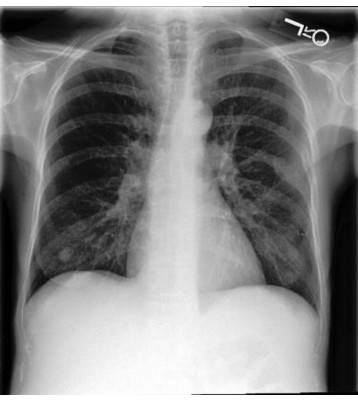

# Medical Report:

No acute cardiopulmonary process. There is no focal consolidation, pleural effusion or pneumothorax. Bilateral nodular opacities that most likely represent nipple shadows. The cardio mediastinal silhouette is normal. Clips project over the left lung, potentially within the breast. The imaged upper abdomen is unremarkable. Chronic deformity of the posterior left sixth and seventh ribs are noted.

(a) CXR  image example 1 from MIMIC dataset

(b) CXR report example 1 from MIMIC dataset

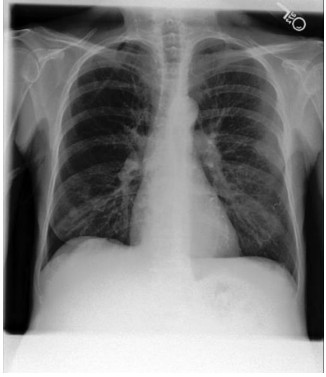

# Medical  Report:

No acute cardiopulmonary abnormality. The cardiac, mediastinal and hilar contours are normal. Pulmonary vasculature is normal. Lungs are clear. No pleural effusion or pneumothorax is present. Multiple clips are again seen projecting over the left breast. Remote left-sided rib fractures are also re-demonstrated.

(a) CXR  image example 2 from MIMIC dataset

(b) CXR report example 2 from MIMIC dataset

Figure 2: CXR dataset examples from MIMIC-CXR.

## A.2 IMAGE-CAPTION DATASET

Our Image-Caption Dataset consists of ROCO and MedICaT. ROCO comprises over 80,000 image-caption pairs. MedICaT includes over 217,000 medical images and their corresponding captions. Figure 5 demonstrates the cases of ROCO.

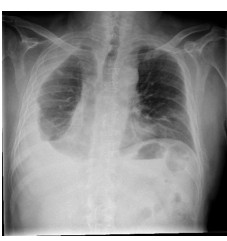

**Medical Report:**
radiografi actual comp con previ 26 juni persistent derram pleural derech . pequen atelectasi subsegmentari bas izquierd . sutur cerclaj esternotomi medi . engros pleural apical izquierd.

**Translated Report:**
Chest X-ray compared to previous one from June 26 shows persistent right pleural effusion, small subsegmental basal atelectasis on the left, and thickening of the left apical pleura. There is a suture cerclage from a previous sternotomy.

**Generated Report:**
The current chest X-ray reveals persistent right pleural effusion, small subsegmental basal atelectasis on the left, and thickening of the left apical pleura. Additionally, there is evidence of a suture cerclage from a previous sternotomy.

(a) CXR image example 1 from PadChest dataset

(b) Spanish report example 1 from PadChest dataset

(c) Translated report example 1 from PadChest dataset

(d) Genrated report example 1 from PadChest dataset

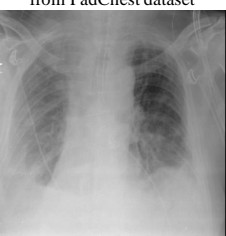

**Medical Report:**
derram pleural bilateral predomini derech . pequen infiltr parch ambos hemitorax . tub endotraqueal . cerclaj esternotomi medi .

**Translated Report:**
Bilateral pleural effusion with predominant right-sided involvement. Small patchy infiltrates in both lung fields. Endotracheal tube in place. Previous sternotomy and mediastinal cerclage.

**Generated Report:**
The chest X-ray shows bilateral pleural effusion with a predominant right-sided involvement. There are small patchy infiltrates in both lung fields. An endotracheal tube is in place, and there is evidence of previous sternotomy and mediastinal cerclage.

(a) CXR image example 2 from PadChest dataset

(b) Spanish report example 2 from PadChest dataset

(c) Translated report example 2 from PadChest dataset

(d) Generated report example 2 from PadChest dataset

Figure 3: CXR dataset examples from PadChest.

## A.3 IMAGE-CLASS DATASET

CXR images and their corresponding labels are also part of our dataset. Figure 6 demonstrates some cases of Image-Class Dataset and the Statistic of diseases in the dataset. Meanwhile, we also illustrate the prompt template used in our pre-training process in Tab A.3

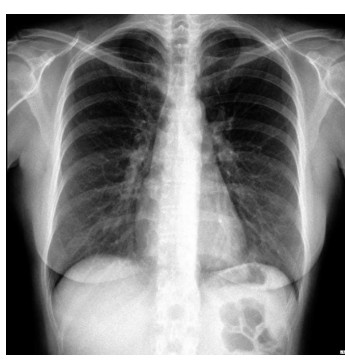

# Medical Report:

No evidence of pleural effusion is observed. No evidence of enlarged cardio mediastinum is observed. No cardiomegaly is identified in the examined region. There are findings suggestive of consolidation..

(a) CXR image example 1 from other dataset

(b) CXR report example 1 from other dataset

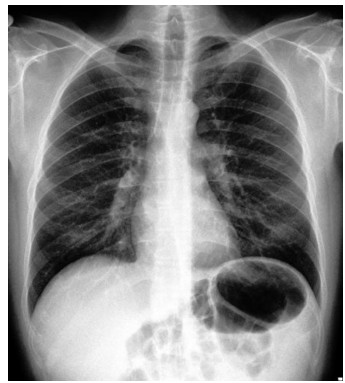

# Medical Report:

No signs of enlarged cardio mediastinum is observed. There is no indications of fracture in the radiograph. No evidence of pleural effusion is observed. The radiograph does not show any signs of cardiomegaly. The radiographic examination of the chest reveals no significant abnormalities or pathologies.

(a) CXR image example 2 from other dataset

(b) CXR report example 2 from other datasets

Figure 4: CXR dataset examples from Other Datasets (Brax, Candidptx, CheXpert, Jfhealthcare, Nih, Vindr).

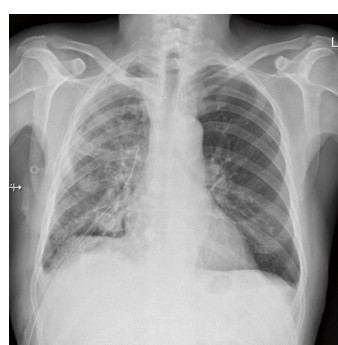

# Medical Caption:

Chest X-ray, posterior-anterior view after the surgical removal of the intermediate lobe of the right lung. Drain in the right pleural cavity. The postoperative chest radiograph revealed no pneumothorax.

(a) CXR image example 1 from ROCO dataset        (b) CXR report example 1 from ROCO dataset

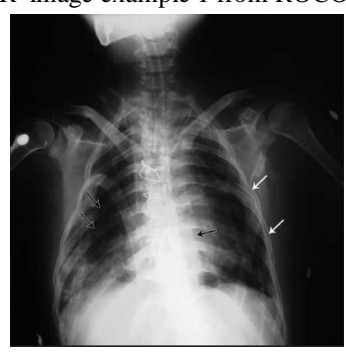

# Medical Caption:

Chest x-ray of the patient (anteroposterior view) shows a small and bell-shaped thoracic cage (white arrows) with a round heart (black arrow in the middle). Thin ribs and slender long bones are also visible (black arrows on the ribs).

(a) CXR image example 2 from ROCO dataset        (b) CXR report example 2 from ROCO dataset

Figure 5: Dataset example from ROCO.

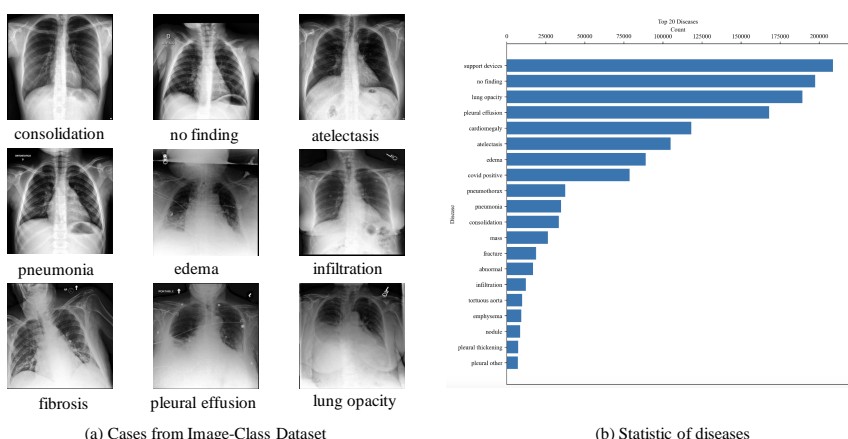

(a) Cases from Image-Class Dataset                    (b) Statistic of diseases

Figure 6: Illustration for Image-Class Datasets.

**Text Prompts**

1. a xray image showing the characteristic signs of
2. a xray depicting the typical features of
3. a detailed xray revealing the bone structure affected by
4. a diagnostic xray highlighting the presence of
5. a xray of the chest with signs of
6. a close-up xray image focusing on
7. a xray of a limb affected by
8. a frontal xray image displaying signs of
9. a xray of the area showing
10. a xray showing an acute case of
11. a xray demonstrating the severity of
12. a digital xray of a patient with
13. a xray highlighting the complications associated with
14. a preoperative xray of a patient diagnosed with
15. a xray showing the unexpected discovery of
16. a routine xray screening that detected
17. a xray with a detailed view of
18. a labeled xray image identifying the areas affected by
19. a targeted xray of the region commonly affected by
20. an underexposed xray of a case of
21. a xray of the barely visible signs of
22. a low resolution xray image of
23. a poorly taken xray of
24. a cropped xray focusing on
25. a xray with the subtle markings of
26. a high-contrast xray of a difficult to detect
27. A brightened xray image of
28. a xray of a pristine
29. a xray of
30. a xray showing the soiled area from
31. a darkened xray revealing
32. a xray of the intriguing
33. a close-up xray of
34. a xray with excellent lighting of
35. a blurry xray of
36. a xray depicting
37. a jpeg corrupted xray image of
38. a xray with a magnified view of
39. a diagnostic xray pinpointing the location of
40. a xray capturing the classic sign of
41. a xray exhibiting the early stages of
42. a bad xray of

43. a xray of the hard to see
44. a low resolution xray of the
45. a bright xray of
46. a dark xray of the
47. a good xray of
48. a xray showcasing the distinct pathology of
49. a high-definition xray image demonstrating the features of
50. a oblique xray view capturing the essence of
51. a comprehensive xray revealing the full extent of
52. a xray snapshot highlighting the critical areas of
53. a medical xray photograph illustrating the anomaly of
54. an advanced xray scan showing the intricate details of
55. a xray image with annotations of the
56. a panoramic xray encompassing the entire scope of
57. a xray with contrast dye emphasizing
58. a xray with highlighted annotations showing
59. a detailed xray mapping the structure compromised by
60. a precise xray pinpointing the origin of
61. a xray with a comparative analysis of
62. a xray with advanced imaging techniques highlighting
63. a follow-up xray indicating the healing progress of
64. a xray showing the differential diagnosis indicators of
65. a post-treatment xray showcasing the resolution of
66. a targeted xray using contrast to delineate
67. a xray with a panoramic view focusing on
68. a xray with a silhouette view of
69. a xray with a spotlight effect on
70. a xray with a windowed view to analyze
71. a xray with a schematic diagram for educational purposes on
72. a teaching xray with labeled structures affected by
73. a xray with a ghosted view to highlight
74. a xray with a false-color enhancement to visualize
75. a xray with an embossed effect to accentuate the texture of
76. a xray with a magnification loupe for close examination of
77. a xray with a highlighted outline of the affected area by
78. a microfocus xray detailing the minute structures within
79. a xray with edge enhancement to clarify the margins of
80. radiographic evidence for
81. signs of
82. convincing signs of
83. focal consolidation concerning for
84. evidence of
85. suggest the presence of
86. convincing evidence of

87. severe
88. These findings would be consistent with
89. a rapidly developing
90. consider worsening
91. worrisome for
92. acute
93. concerning for
94. the possibility of supervening  would have to be considered in the appropriate clinical setting
95. patient was discharged from ED with diagnosis of
96. should also be considered
97. in the appropriate clinical setting  should be considered
98. developing
99. findings may be due to
100. consistent with
101. but in the right clinical setting could be due to
102. should also be considered
103. an early focus of
104. findings to suggest
105. there is good evidence for
106. concerning for early developing
107. includes  in the appropriate clinical setting
108. an early  should also be considered
109. appears more likely
110. suggestive of
111. patient presents with
112. suspicion of
113. Diagnosis:
114. the findings could correspond to a radiological
115. possible radiological
116. these findings suggest the possibility of

### A.4 IMAGE-SEGMENT DATASET

CheXlocalize and ChestX-ray14 constucts our Image-Segment dataset. For unified training, we convert the coordinates of disease to segmentation mask while use different color to represent different diseases.

---

**Algorithm 1** Training algorithm of *UmiF*.

---

**Dataset**: Image-Report Dataset $\mathcal{D}^{ir}$, Image-Cation Dataset $\mathcal{D}^{ica}$, Image-Class Dataset $\mathcal{D}^{icl}$, Image-Segment Dataset $\mathcal{D}^{ics}$

**Vision Encoder**: student network $E$, teacher network $\overline{E}$,

**for** each updating step **do**

    Sample 4 tasks from the task set {Image-Report, Image-Caption, Image-Class, Image-Segment} with replacement according to the dataset size (larger dataset has higher probability to be sampled)

    For each sampled task, randomly select M pairs in the corresponding dataset, resulting in a batch $\{(i, s)\}$ with 4M image-supervision pairs.

    **for** each pair in the batch **do**

        Tokenizer the image-supervision pair and obtain $\mathbf{X}^i$ and $\mathbf{X}^s$ in student and teacher model

        Apply token gouping strategy in student model and get $\mathbf{X0}$, $\mathbf{X1}$

        $\mathbf{f0}$, $\mathbf{f1} = E(\mathbf{X0}), E(\mathbf{X1})$

        $\mathbf{t} = \overline{E}(Concat(\mathbf{X}^i, \mathbf{X}^s))$

    **end for**

    Gather $\mathbf{f0}$, $\mathbf{f1}$, $\mathbf{t}$ in the current batch (4M in total)

    Compute $\mathcal{L}_{align}$ and $\mathcal{L}_{con}$

    Update the model with $\mathcal{L}_{align} + \mathcal{L}_{con}$

**end for**

---

