# OpenReview forum: "Learning Robust Representations for Medical Images via Unifying (Self-)Supervisions"
_ICLR.cc/2025/Conference — Submitted to ICLR 2025_

### Official Review · Reviewer_Mh81 · 2024-10-29

**Soundness:** 3
**Presentation:** 3
**Contribution:** 3
**Rating:** 6
**Confidence:** 4

**Summary:**

This paper propose a new unified pre-training framework to pre-train the model on medical images by unifying all common types of supervisions. They first convert all input data into token embeddings including image and language, segmentation (depend on the task of the dataset) modalities. Then, they apply a flexible grouping strategy that split token embeddings into two groups before feeding these two groups into a VIT and consider it as positive pairs for contrastive learning task. They also apply mask modeling on these two groups. They collect 1.66M samples from 14 public datasets and pre-train on this dataset. They conduct experiments on many downstream tasks.

**Strengths:**

- They collected a large scale medical dataset from many supervision tasks for pre-training purposes, which are all public datasets.
- I like the idea and method of tokenizing images, text, etc. to unify all common types of supervisions into a pre-training framework for medical encoders.
- Various experiments are conducted to verify whether the new pre-training framework is good.

**Weaknesses:**

- There are some common self-supervised pre-training methods that you need to compare your method with to make sure your pre-training framework on medical images is strong because these methods can also utilize images in many datasets from different tasks and consider images and corresponding augmentation version as positive pairs before using contrastive algorithms such as InfoNCE, Graph-Matching, etc. to pre-train the model (instead of using two groups of token embeddings obtained from flexible grouping strategy as positive pairs for contrastive algorithm as in your method). For example:
  - Duy Minh Ho Nguyen et al. LVM-Med: Learning Large-Scale Self-Supervised Vision Models for Medical Imaging via Second-order Graph Matching. In NeurIPS, 2023
  - Adrien Bardes et al. Vicregl: Self-supervised learning of local visual features. In NeurIPS, 2022.
  - Mathilde Caron et al. Emerging properties in self-supervised vision transformers. In Proceedings of the IEEE/CVF international conference on computer vision, 2021
- The performance on downstream tasks are good but I think performance on classification and segmentation are not improved enough compared to Med-UniC (VIT-B). Can you give me an analysis of where and why the method shows improvements (or doesn't) compared to baselines ? It could provide more insight into the strengths and limitations of your method.

**Questions:**

Please address the comment in weakness part.

---

### Official Review · Reviewer_q7Gu · 2024-10-30

**Soundness:** 2
**Presentation:** 1
**Contribution:** 3
**Rating:** 3
**Confidence:** 4

**Summary:**

The Authors propose UmiF, a framework that aims to unify pre-training of any arbitrary Image-Label pair to train a robust encoder for any modality. To achieve this the authors propose three tokenizers, that each allows embedding to a shared unified token space (One Image, One Text, One Image Labels/Segmentation). These image / label tokens are either left split or merged to a certain degree, before being used for SSL training through contrastive training in a CLIP'esque fashion or through a reconstruction task.
They train their model on a wide variety of paired pre-training datasets and evaluate it on a broad set of downstream tasks, highlighting the final performance of UmiF's method.

While innovative, the experiments are insufficient to highlight the proposed methodology. The authors stack a) a larger pre-training dataset b) Token mixing c) Masked and Contrastive losses together and don't provide experiments that disentangle which part brings performance and which part does not. Moreover, the presentation and language used in this paper are of insufficient quality and need a lot of work.

**Strengths:**

The proposal to not only include Image-Report pairs but also Images with other Supervision signals is an interesting and to my knowledge novel premise for the medical domain. Their idea of mixing image and supervision tokens are also innovative.
Moreover, the amount of experiments conducted is broad, highlighting the generality of the ViT feature extractor.

**Weaknesses:**

While the premises are highly interesting the paper in it's current form has some issues:
1. **Stacking of contributions** Currently the authors stack a variety of things on-top and don't ablate it properly, namely a) a larger pre-training dataset b) the token mixing block and c) the multiple SSL losses. Currently, it is impossible for a reader to know if their methodology is better downstream than the competing methods, as they create a much larger training data corpus. Maybe it's the masked reconstruction component, maybe not.
2. **Presentation** The presentation of the paper as of right now is poor. It was very hard to read, as the language quality leaves a lot of room for improvement and should be checked by an English speaker to rework the manuscript. Moreover, Fig. 1 does a bad job of explaining the method contributing to difficulties in understanding the proposed method. Figures in the appendix are badly presented: Fig. 2 text is way too large.  Table 3 does not have a caption, The description of the Image-Segment Dataset (Section A.4) is basically non-existent and should be filled accordingly.
3. **Reproducibility** Currently the author's don't sufficiently explain their configuration of their methods. How was _r_ chosen ? In the text it is mentioned sometimes 1 and sometimes random between [0,1]. How did the authors split their data? Was there a train-test split during pre-training and fine-tuning? Was there different weightings between the losses?

### Minor Points
- The ablation of _r_ values does not contain r of 0.9 and 1.0 - It's mentioned in the text that these performed substantially worse, but I would like to have these values included in the table. Moreover this table should provide not only RSNA 1% AUC values. The downstream adaptation are just learning of a linear-layer so please provide ablations on more datasets and all values to show if the mixing of tokens actually provides a benefit.
- The distinction between what this paper does relative to other paper's feels not well worked out. It would help a lot to see what makes this work distinct.
- Similarities to MedUniC Paper. This paper's Table 2 is very similar to their Table 2 -- I would prefer to highlight this in the caption.
- There are so many typos in this manuscripts. E.g. spellings of baseline methods:  MedKLIIP/MedKILP/MedKLIP. It feels like no one proof-read this paper ever.
- The Algorithm 1 is way too text heavy. If the authors want to go into detail about the sampling of their datasets they should move this into a separate algorithm to keep readability high.
- The authors mention the importance of sampling smaller dataset more regularly but provide no results. Would be great to see an ablation table on this claim in the appendix.

**Questions:**

- Q1: Did you ablate the importance of using the masking and the contrastive loss by themselves?
- Q2: How was _r_ chosen?
- Q3: Shouldn't r be symmetric around 0.5? r=0 would just flip image embeddings to supervision and the other way around. Is this correct, or if not why not?
- Q4: Have you tried replacing the mixing with a standard masking/token drop-out layer? Would be interesting to see if one actually has to mix tokens or if the dropping of tokens provides a similar regularizing effect.

---

### Official Review · Reviewer_v5p5 · 2024-10-31

**Soundness:** 2
**Presentation:** 1
**Contribution:** 2
**Rating:** 5
**Confidence:** 3

**Summary:**

This paper introduces a multi-supervision unification strategy for medical image pretraining. The method allows report, segmentation, and classification (+ some others) types of supervision to jointly train one representation. The used modality is Chest X-ray (CXR). The authors collected a large-scale dataset sourced from the public domain, reaching 1M images and 1.66M supervision labels. It is reported the model, namely, UmiF reaches SOTA for a number of downstream tasks.

**Strengths:**

1. The authors have gone through tremendous effort in collecting and inventorying the datasets. I can imagine the implementation wouldn't be easy to iron out the differences in the datasets and put them together to train one model. For that, I believe the significance of the paper should be pointed out.

2. the benchmarking is comprehensive, ranging over the common medical image analysis tasks.

**Weaknesses:**

1. The clarity of the paper is a concern. Many places in the text lack proper explanation and are somewhat confusing.  For example, L230 "This interesting design allows more diverse views and enriches the learning tasks with many possibilities, surpassing previous VL learning approaches.", The authors should clearly state why it is interesting. What are the many possibilities? What are the other diverse views (isn't the modality just CXR)? What evidence indicates your method surpasses the previous VL learning approach?
2. Furthermore, Sec 3.2, perhaps the most important section in the paper is not well written, I've read it a few times and I still don't believe I have grasped the exact approach.
3. I find Figure 1 hard to follow, the quantities in Sec 3.2 should be mapped to the figure. I also don't get the colour coding in Figure 1 for those tokens. The yellow/blue/no boundary cubes are also a very confusing way of presentation.
4. The improvement over the previous state-of-the-art is marginal around 1 point in various measurements. As the authors claim a large-scale dataset of 1.66M image-supervision pairs vs "previous effort of mostly limited to 380K image-report pairs or 838K images", it is worth rethinking whether the effort spent on training such a large model on the twice amount of data makes sense.
5. The title claims "learning robust representation for medical images ...", medical images are not just CXR, I would recommend claiming a lesser scope unless common modalities such as MRI/CT are also used.
6. In Sec 3.1, the authors use "modality abstraction", which sounds cool but I would say it is actually confusing, the procedure is a label format conversion.

**Questions:**

Please address weaknesses #2&4. The paper could use some professional editing services.

---

### Official Review · Reviewer_6fLQ · 2024-11-03

**Soundness:** 3
**Presentation:** 2
**Contribution:** 2
**Rating:** 6
**Confidence:** 4

**Summary:**

The paper presents a pre-training framework named UmiF (Unified Medical Image Framework), trained with diverse supervision types, such as self-supervision and external supervision (e.g., segmentation annotations, text descriptions), aiming to create robust, task-agnostic representations for medical images. UmiF converts various supervision inputs into token embeddings, utilizing a unified token space and flexible token grouping for contrastive learning and mask modeling. The pre-trained model yields state-of-the-art performance on several downstream tasks, including classification, segmentation, and visual question answering.

**Strengths:**

- The framework combines multiple types of supervision (self-supervision, segmentation, and textual descriptions), which allows UmiF to accommodate and generalize well across a variety of medical image tasks.

- By designing a unified token space and a novel flexible token grouping strategy, the authors effectively manage diverse data sources, which is essential given the limited size and annotation diversity in medical datasets.

- The study is based on a large pre-training dataset of 1.66 million samples from 14 public datasets.

**Weaknesses:**

- UmiF is trained solely on publicly available datasets, and it is primarily focused on 2D X-ray images. What about the generalizability of UmiF to other medical imaging modalities like CT or MRI? And what about the generalizability to domain shift problem due to differences in patient populations and equipment quality.

- There is a paper has the similar motivation. This paper unifies different data sources by homogenizing every supported input and output (including image, language, segmentation, bounding box…) into a sequence of discrete vocabulary tokens. However, this paper is not cited and compared in related work as well as experimental sections.
[1*] Lu J, Clark C, Zellers R, et al. Unified-io: A unified model for vision, language, and multi-modal tasks[C]//The Eleventh International Conference on Learning Representations. 2022.

- The motivation of flexible token grouping strategy is missing. I am wondering how the authors came up with this method to unify tokens from diverse data sources.

- In Table 6, it is not clear why r=0.7 also shows a very good performance. According to other results of r>0.2, the tendency is the performance decreases with the increasing of r. Moreover, it is not clear why r>0.8 will lead to large performance descent.

- In Table 7, it is not clear why image-segment shows the worst performance in most cases. Is it because tokens of segment is not good to represent the segment?

- No ablation study of two losses.

- Some typos, such as Tab 5 (but the authors used Table 4, 6 etc.). In line 511, one reference is ‘?’

**Questions:**

See section of Weaknesses

---

### Official Review · Reviewer_xsr1 · 2024-11-06

**Soundness:** 2
**Presentation:** 2
**Contribution:** 2
**Rating:** 3
**Confidence:** 5

**Summary:**

This paper presents UmiF, a pre-training framework for medical image encoders that integrates multiple types of supervision, including self-supervision and annotations like segmentation labels, into a unified approach. UmiF creates a common embedding space with a token grouping strategy to leverage diverse data types for various downstream tasks. Pre-trained on 1.66 million samples from 14 public datasets, UmiF was evaluated in classification, segmentation, detection, retrieval, and VQA tasks.

**Strengths:**

- The paper introduces unifying representations from multiple supervisions into a single embedding space for self-supervised learning and proposes a grouping strategy for mixed learning of representation vectors.

- The model’s effectiveness is validated through evaluations across four different downstream tasks.

**Weaknesses:**

- **Contrastive Learning Design Concerns**: The design of the contrastive learning setup after grouping raises questions. According to the paper, a positive pair is represented by \(f1_i, f0_j\) where i and j are indices from different data points, meaning \(f1_i\) and \(f0_j\) are from different samples. Typically, a positive pair should be \(f1_i, f0_i\), where both elements come from the same sample, making the current approach unclear.

- **Unfair Comparisons in Downstream Tasks**: There are substantial fairness issues in the downstream task comparisons. Competing models, such as Med-Unic and MGCA, are pre-trained on datasets with 380K and 217K samples respectively, whereas this study uses 1.66 million data pairs, including 1 million images. The model’s performance advantage in downstream tasks may stem from this large data disparity, making it difficult to attribute improvements solely to the proposed pre-training strategy.

- **Performance in Table 2**: In Table 2, despite using more training data and supervision than Med-Unic, the proposed model does not achieve the best performance, which raises questions about the efficiency of the approach.

- **Limited Ablation Study on Parameter r**: In the ablation study on the parameter r, only 1% of the RSNA dataset is used, rather than the full dataset, and no similar experiments are conducted on other datasets. It is unclear if the chosen r value on RSNA is robust and generalizable to other tasks, as this limited evaluation does not provide strong evidence of robustness.

- **Inconsistencies Between Text and Figures**: There are inconsistencies between the text and figures. For instance, the text describes vector groups as Group 1 and Group 0, but the figure labels them as Group 1 and Group 2.

- **CLS Token Generation Unclear**: The generation of the CLS token information is not clearly explained.  According to the figure, the CLS token appears to be an output of the Flexible Token Grouping, but the paper does not specify how the CLS token is produced. Further clarification on this process would improve understanding.

**Questions:**

please refer to Weaknesses

---

### Meta-Review · Area_Chair_Fh9x · 2024-12-17

**Metareview:**

The paper proposes UmiF, a pre-training framework for medical image encoders that unifies multiple types of supervision, including self-supervision and annotations like segmentation labels. A unified token space and a flexible token grouping strategy are introduced to handle diverse data sources effectively, with the goal of improving generalisation across medical imaging tasks. Reviewers appreciated the effort in integrating various supervision signals and the significant work in collecting and curating datasets.

However, reviewers unanimously raised concerns about several aspects: the design of the contrastive learning component, unfair comparisons in downstream tasks, the lack of thorough ablation studies on r, and inconsistencies between the text and figures. The clarity of presentation was also noted as a limiting factor. With the majority of reviews expressing negative opinions and no rebuttal provided to address these valid concerns, I recommend rejection.

**Additional Comments On Reviewer Discussion:**

There is no rebuttal and thus no discussion.

---

### Decision · Program_Chairs · 2025-01-22

Reject